# Machine Learning in Predicting Tooth Loss: A Systematic Review and Risk of Bias Assessment

**DOI:** 10.3390/jpm12101682

**Published:** 2022-10-09

**Authors:** Akira Hasuike, Taito Watanabe, Shin Wakuda, Keisuke Kogure, Ryo Yanagiya, Kevin M. Byrd, Shuichi Sato

**Affiliations:** 1Department of Periodontology, Nihon University School of Dentistry, Tokyo 101-8310, Japan; 2Department of Innovative Medical Science, Tokai University School of Medicine, Isehara 259-1193, Japan; 3Dental Research Center, Nihon University School of Dentistry, Tokyo 101-8310, Japan; 4Division of Applied Oral Sciences, Nihon University Graduate School of Dentistry, Tokyo 101-8310, Japan; 5Department of Neurology, Hematology, Diabetology, Endocrinology, and Metabolism (3rd Department of Internal Medicine), Faculty of Medicine, Yamagata University, Yamagata 990-0021, Japan; 6Lab of Oral & Craniofacial Innovation (LOCI), ADA Science & Research Institute, Gaithersburg, MD 20879, USA

**Keywords:** machine learning, boosting, deep learning, tooth loss, periodontitis, prognosis

## Abstract

Predicting tooth loss is a persistent clinical challenge in the 21st century. While an emerging field in dentistry, computational solutions that employ machine learning are promising for enhancing clinical outcomes, including the chairside prognostication of tooth loss. We aimed to evaluate the risk of bias in prognostic prediction models of tooth loss that use machine learning. To do this, literature was searched in two electronic databases (MEDLINE via PubMed; Google Scholar) for studies that reported the accuracy or area under the curve (AUC) of prediction models. AUC measures the entire two-dimensional area underneath the entire receiver operating characteristic (ROC) curves. AUC provides an aggregate measure of performance across all possible classification thresholds. Although both development and validation were included in this review, studies that did not assess the accuracy or validation of boosting models (AdaBoosting, Gradient-boosting decision tree, XGBoost, LightGBM, CatBoost) were excluded. Five studies met criteria for inclusion and revealed high accuracy; however, models displayed a high risk of bias. Importantly, patient-level assessments combined with socioeconomic predictors performed better than clinical predictors alone. While there are current limitations, machine-learning-assisted models for tooth loss may enhance prognostication accuracy in combination with clinical and patient metadata in the future.

## 1. Introduction

Predicting whether compromised teeth can be retained over the long term is a crucial component of treatment planning for oral health care. Tooth loss can generally be prevented if disease is diagnosed and treated at an early stage for both caries and periodontal disease. However, while current periodontal disease prognostic systems are reported to have some success and reproducibility in predicting tooth loss and tooth retention, low sensitivity defines these models [1]. Thus, it is essential to develop better prognostication tools based on tooth mortality, particularly in the field of periodontology.

Historically, prognosis has included assessment at the operator level. For example, in 1978 Hirschfeld and Wasserman published a record of periodontal maintenance in 600 patients with 15,666 teeth for an average of 22 years [2]. They classified a tooth as “Questionable” if it was found to have furcation defects, non-healing periodontal pockets, extensive alveolar bone resorption, and significant mobility associated with deep periodontal pockets. The authors reported that the loss rate of teeth classified as “Questionable” was 31.3%. This study was likely the first to report the prognosis of periodontally compromised teeth.

In 1991, McGuire proposed a system for determining the prognosis of teeth using prognostic factors commonly used in periodontal treatment [3]. The authors classified teeth into five categories: Good, Fair, Poor, Questionable, and Hopeless. This system was applied to the data of 2484 teeth from 100 patients who had been under maintenance. The results showed that the teeth that were judged to have a “Good” prognosis at the initial visit were still judged to be “Good” at 5 and 8 years, with a probability of more than 80%, but other judgments showed a low accuracy rate. Ultimately, the authors proposed that “if the initial prognostic value is not Good, it would be easier and more accurate to determine the prognostic value by coin toss”.

Although many prognostication systems have been proposed since these publications [4,5,6], selecting teeth to be extracted and making prognoses remain challenging in the 21st century, and these systems are not commonly used in daily practice. This remains an unmet need for both patients and oral health care workers, especially as we enter the era of precision oral medicine [7]. In personalized and precision medicine, patients are stratified based on their disease subtype, risk, prognosis, or treatment response [8]. For implementation of these strategies in the dental clinic, there is an urgent need to enhance prognostication models for many oral diseases and conditions—including tooth loss.

Computational methods such as artificial intelligence and machine learning are emerging in oral health care to solve these diagnostic and prognostic challenges [9]. Machine learning, which is a subset of artificial intelligence, refers to computationally intensive methods that use data-driven approaches to develop models that require fewer modeling decisions by the modeler than traditional modeling techniques [10]. Currently, machine learning is widely accepted due to its ability to develop prediction models, including offering more flexible modeling and its ability to analyze ‘big’, non-linear, and high dimensional data as well as to model complex clinical scenarios [11]. These approaches can be efficiently tested in healthcare applications, such as disease diagnosis, medical image analysis, big data collection, research and clinical trials, management of smart health records, and prediction of disease outbreaks [12]. The proposal of new prognostication systems using machine learning is expected to increase in the near future; therefore, it is necessary to critically evaluate these emerging methods.

Using a prediction model considered to have a high risk of bias may lead to unnecessary or insufficient interventions. Rigorous risk of bias evaluation is therefore essential to ensure the reliable application of prognostication models. Here, we aimed to evaluate the risk of bias in prognostic prediction models of tooth loss that use machine learning.

## 2. Materials and Methods

### 2.1. Focused Question

This study was conducted using the Preferred Reporting Items for Systematic Review and Meta-analysis (PRISMA) guidelines. Our focused question was constructed according to the Participants Intervention Comparison Outcome and Study (PICOS) strategy.

Population: Adult patients.

Intervention/Comparison: Machine learning models applied in prognostic prediction models for tooth loss. Teeth level or patient level.

Outcome: Analysis of machine learning performance and validation assessed by accuracy or area under the curve (AUC),

Study design type: Prediction model studies based on a cross-sectional, case-control, or prospective design.

### 2.2. Search Strategy

The published literature was searched in two electronic databases (MEDLINE via PubMed and Google Scholar). Boosting algorithms represent one of the most promising methodological approaches to data analysis. Currently, gradient boosting algorithms are primarily used as powerful ensemble machine learning algorithms in healthcare fields, such as the extreme gradient boosting decision tree (XGB), light gradient boosting machine (LightGBM), and categorical boosting (CatBoost). XGBoost was introduced in 2016, LightGBM in 2017, and CatBoost in 2019 [13]. Therefore, it is important to search for relevant recent publications, and the present study was restricted to studies published after 2012. The keywords used in the MEDLINE search were as follows: (“Humans”[Mesh]) AND (((“Algorithms”[MeSH Terms] OR “decision support systems, clinical”[MeSH Terms] OR “models, dental”[MeSH Terms] OR “models, theoretical”[MeSH Terms]) AND (“Tooth Loss”[MeSH Terms] OR “Tooth Extraction”[MeSH Terms]))). The keywords used in the Google Scholar search were as follows: (boosting OR boosted OR GBT OR GBM OR “random forest”) AND (“Tooth Loss” OR “Tooth Extraction”). The search process planning and all electronic searches were conducted by one examiner (A.H.), with the cooperation of the healthcare librarian. In addition, the reference lists of each included study were checked manually by two examiners (T.W. and S.W.) for possible additions.

### 2.3. Inclusion and Exclusion Selection Criteria

Only the prediction model studies based on a cross-sectional, case-control, prospective design (including case series) or retrospective design were included. Studies that developed prediction models using primary or secondary data (registry and electronic health records) were also included. Reviews, letters to the editor, and clinical guidelines were excluded from the review. The participants were adults aged 18 years or older. The predictors included any dental, medical, or social measures, regardless of how they were determined. Although both development and validation were included, studies that did not assess the accuracy or validation of boosting models (AdaBoosting, gradient-boosting decision tree, XGBoost, LightGBM, CatBoost) were excluded. The outcome was tooth loss or prognosis of teeth at the teeth or patient levels. Studies that reported the accuracy or AUC of the prediction model were included. AUC measures the entire two-dimensional area underneath the entire receiver operating characteristic (ROC) curve. ROC curves is a graph showing the performance of a classification model at all classification thresholds. AUC provides an aggregate measure of performance across all possible classification thresholds. The higher the AUC, the better the model is at distinguishing between patients with the disease and no disease. The titles and abstracts of all potential publications were independently screened by two reviewers (T.W. and S.W.) against the inclusion criteria, and the article’s eligibility was confirmed after discussion. In cases of disagreement, an independent reviewer (A.H.) was consulted.

### 2.4. Data Extraction and Risk of Bias Assessment

For studies that fulfilled the inclusion criteria, two authors (T.W. and S.W.) independently extracted bibliographic details regarding the patients, predictors, outcomes, and analysis. Furthermore, the AUC and accuracy of the primary assessment in each study were also extracted. Because of the heterogeneity of model development and validation studies, a meta-analysis of their diagnostic performance was not appropriate. Rather, a narrative synthesis of evidence was preferred. The included studies were critically appraised using the Prediction model Risk Of Bias ASsessment Tool (PROBAST) [14]. PROBAST assessment was applied to the most developed model in each study. Two investigators (A.H. and T.W.) independently assessed the risk of bias in the included studies. PROBAST examines the extent to which a model’s risk predictions are likely to be accurate when applied to new individuals and depends on four domains: participants, predictors, outcomes, and analysis. Each domain included signaling questions (two for participants, three for predictors, six for outcome, and nine for analysis) to aid in judging the risk of bias. The results are reported in tables and figures presenting the key methodological features and main findings of all the included studies and the risk of bias assessment.

## 3. Results

### 3.1. Search Results

The article selection process is presented in Figure 1. The initial electronic search retrieved 1579 articles. Independent scrutiny of titles and abstracts identified seven potentially relevant studies for full-text review. After full-text evaluation, two articles did not meet our inclusion criteria and were excluded [15,16]. Both the excluded studies made predictions without boosting the algorithm. Finally, five studies formed the basis of this study [17,18,19,20,21].

### 3.2. General Study Characteristics and Results

A summary of the general information of each study is presented in Table 1. Four of the five included studies were development/validation studies [17,18,19,20]. Another study was a development study [21]. The included studies were published between 2019 and 2022. The number of patients included in each study ranged from 94 to 11,977. Of the four validation studies, two studies used “Hold-out validation” [17,18], and one study used “10-fold cross validation” [19]. All studies employed oral variables as predictors. Furthermore, two studies used socioeconomic variables [19,20]. Three out of the five studies included predictions made at the tooth level [17,18,21], and the two other studies with socioeconomic variables made predictions at the patient level [19,20]. All five included studies employed XGB as the model algorithm, and four employed random forest optimization (RFO) [17,19,20,21]. Three studies employed decision tree classification (DTC), including recursive partitioning [17], classification and regression trees [18], and decision tree classifiers [21]. Two studies employed LightGBM [18,20]. All the included studies assessed model accuracy, and four out of five studies also reported AUC [17,18,19,20]. Furthermore, three studies reported F1 [18,19,20], and one study reported the no-information rate [17].

The results of the assessment and validation of each model are presented in Table 2. Each modeling showed high AUC and accuracy, with values larger than 0.8. In three out of four validation studies, XGB outperformed the other models [18,19,20]. However, another study with 11,651 teeth showed no superiority in complex models (RFO, XGB) over simpler models, such as logistic recession (logR) and DTC [17]. A development study with only 94 patients showed the lowest accuracy values (model A: 0.689 for XGB, 0.8312 for RFO, and 0.8413 for DTC) [21].

### 3.3. Risk of Bias Assessment

A summary of the PROBAST assessments is shown in Figure 2. As all included studies employed XGB, the PROBAST assessment was conducted for the XGB model in all studies. All the studies were relevant to the review question and had a high risk of bias. In the patient domain, a development study with 96 patients was judged as having a high risk of bias [21]. There were no clear inclusion or exclusion criteria for this study. In the predictor domain, two studies were judged to be at a high risk of bias. In a validation study with 11,651 teeth, no radiographs were available to avoid repeated radiographic assessment [17]. In the development study, since assessments of predictors and outcomes were conducted simultaneously, both predictor and outcome domains were ranked as having a high risk of bias [21]. In the outcome domain, all of the included studies were judged as having a high risk of bias because the decision of tooth extraction would not be made in the standard way, and outcomes were determined using information regarding predictors such as mobility or bone loss. In the analysis domain, all the studies were judged to have a high risk of bias. All validation studies were ranked as having a high risk of bias in this domain because the number of participants with the outcome was quite small [17,18,19,20]. The analysis of predictor selection also influenced this domain. Only two studies accounted for optimism in the models [18,19]. Furthermore, only two validation studies used internal validation techniques to account for any optimism in model fitting [19,20].

## 4. Discussion

In this study, we assessed the risk of bias in teeth prognostic model development and validation studies that applied machine learning methods. The included studies mostly showed high AUC and accuracy values using machine learning modeling. However, all included studies were assigned a score of high risk of bias because of methodological limitations during outcome assessment and data analysis. There is a strong need for modifications and improvements of study methods to enhance the reliability of machine learning algorithms to predict tooth loss.

The signaling questions judged as having a high risk of bias in PROBAST would be distinguishable between unavoidable limitations that are inherently included in tooth extraction decision-making and methodological failures which could be improved. PROBAST was primarily designed for regression-based prediction in medical fields. Thus, signaling questions are not fully applicable to machine learning-based prediction model studies, especially in dentistry. In the outcome domain, PROBAST recommends not to use subjective outcomes. However, for tooth loss, it is challenging to set pre-specified or standardized objective outcomes because tooth loss has more than one etiology. This is one of the limitations inherent to this study. Furthermore, it is also recommended to exclude predictors from the outcome definition. In this clinical question, some clinical variables such as bone loss or mobility may be used in tooth extraction decision-making. Thus, it remains challenging to judge outcomes without knowledge of the clinical variables. This is also a limitation that is inherent to this clinical question. Furthermore, in the modern era, the incidence of tooth extraction is quite low, especially at the tooth level. However, the development study set prognostic categories as outcomes rather than tooth loss [17]. In this case, the influence of the number of events could be avoided. Considering these inevitable limitations, one would rescore one validation study with patient-level outcomes as a low risk of bias [15].

In PROBAST, predictors should be selected based on multivariate modeling. A validation study with 26,005 teeth employed recursive feature elimination to discard predictors that were weakly related to tooth extraction [14]. Another validation study with 19,407 patients used the random forest-based Boruta feature selection algorithm to select only the most relevant predictors for the model [15]. The other three studies selected predictors in an experience-based manner. In future research, it will be necessary to select predictors objectively. As mentioned above, it is inevitable to have an influence of small incidence rate in tooth loss assessment. Thus, researchers must employ analytic frameworks, such as oversampling. One validation study performed random oversampling of the minority class to obtain class balance [15]. Furthermore, in validation studies, it is important to focus on the effects of overfitting. To reduce this influence, it is common to employ a robust measure such as “cross-validation”. K-fold cross-validation is commonly used to assess the validation of machine learning models. This could be achieved by splitting the data into k groups; each unique group is held out as test data, while the remaining k-1 groups are used as training data. In only two validation studies, 10-fold cross-validation was used for model evaluation [15,16]. Currently, the PROBAST guideline tailored for AI (PROBAST-AI) is updating [22]. In the future, clear navigation of validation methods would be available. It would be recommended to follow these guidelines and conduct appropriate tuning. In the present review, a development study without validation analysis was also included [21]. In this study, the inclusion criteria of study participants were not well defined. Furthermore, the prognosis was decided with tooth-related factors which were also included as predictors. Considering these points, this study was judged as “critically high risk of bias”. If we restrict validation studies for inclusion, the overall risk of bias of prognostic prediction models would be judged as moderate to low.

The included studies were roughly categorized into two groups: (1) studies with tooth-level outcomes based on oral predictors [13,14,17] and (2) studies with patient-level outcomes based on oral and socioeconomic predictors [15,16]. In the dental field, especially periodontology, prognostic systems have been developed based on teeth-related and periodontal predictors. However, to date, no highly sensitive prognostic system has been established [1]. Even with the support of machine learning methods, higher AUC and accuracy were not obtained compared with simpler methods [13]. On the other hand, in both patient-level studies, relatively high AUC and accuracy were obtained [15,16]. Furthermore, socioeconomic factors, such as income, education, and employment, were found to be important predictors of tooth loss in these two studies. These findings suggest that the machine learning algorithm models using clinical and patient metadata, such as socioeconomic characteristics, self-reported dental care, and medical conditions will enhance tooth loss prediction compared to dental indicators alone.

We acknowledge several limitations of the present review. First, all four validation studies employed tooth extraction for any reason as the outcome. Using an inconsistent data set would be attributed to a high risk of bias in a systematic review. Studies with tooth extraction for specific reasons, such as periodontal, endodontic, and prosthetic problems would be needed as future studies. Second, we searched only two major literature databases, MEDLINE and Google Scholar, for development and validation studies written in English. There are some possibilities that we may have missed eligible publications. In future systematic reviews, broad literature search methods should be employed.

## 5. Conclusions

This review summarizes existing evidence on the development and validation of tooth retention/extraction prognosis assisted by machine learning models. Although only five studies met our inclusion criteria currently, the number of studies will likely increase in the near future. Although these models showed high AUC and accuracy, they were broadly assessed to have a high risk of bias. Although there are clinical questions related to unavoidable limitations, particular attention and methodological guidance are needed to improve the quality of machine-learning-based clinical prediction models. It is also suggested that the machine-learning algorithm models for patient-level assessment with socioeconomic predictors performed better than tooth-level assessment relying on clinical dental predictors alone. Researchers should modify and improve study methods to enhance the reliability of machine learning algorithms to predict tooth loss.

## Figures and Tables

**Figure 1 jpm-12-01682-f001:**
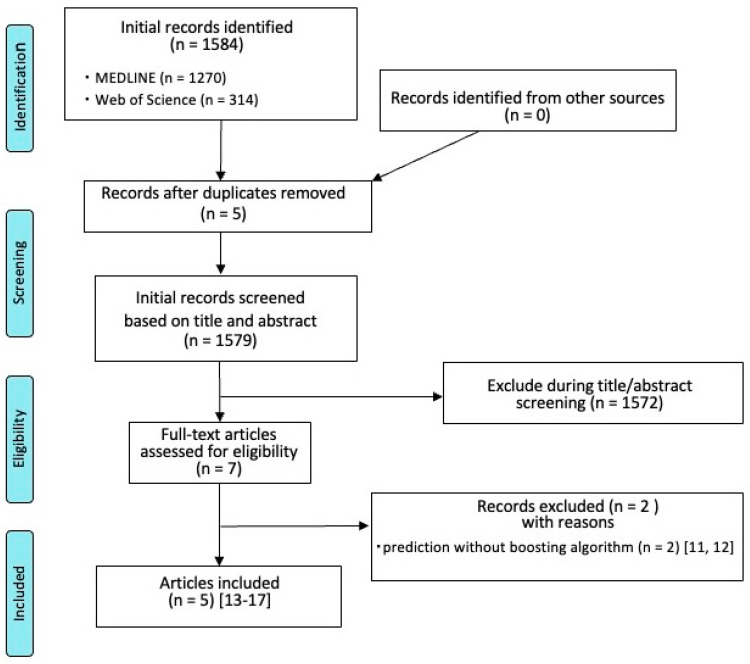
Flow diagram for the selection of studies.

**Figure 2 jpm-12-01682-f002:**
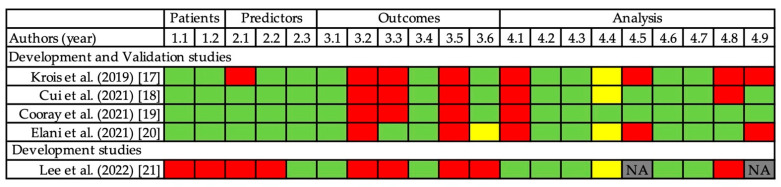
Risk-of-bias assessment with PROBAST signaling questions in four domains [14]. Low (**green**), high (**red**), unclear risk of bias (**yellow**), and not applicable (NA). PROBAST items: 1.1. Were appropriate data sources used? 1.2. Were all inclusions and exclusions of participants appropriate? 2.1. Were predictors defined and assessed in a similar way for all participants? 2.2. Were predictor assessments made without knowledge of outcome data? 2.3. Are all predictors available at the time the model is intended to be used? 3.1. Was the outcome determined appropriately? 3.2. Was a prespecified or standard outcome definition used? 3.3. Were predictors excluded from the outcome definition? 3.4. Was the outcome defined and determined in a similar way for all participants? 3.5. Was the outcome determined without knowledge of predictor information? 3.6. Was the time interval between predictor assessment and outcome determination appropriate? 4.1. Were there a reasonable number of participants with the outcome? 4.2. Were continuous and categorical predictors handled appropriately? 4.3. Were all enrolled participants included in the analysis? 4.4. Were participants with missing data handled appropriately? 4.5. Was the selection of predictors based on univariable analysis avoided? 4.6. Were complexities in the data accounted for appropriately? 4.7. Were relevant model performance measures evaluated appropriately? 4.8. Were model overfitting and optimism in model performance accounted for? 4.9. Do predictors and their assigned weights in the final model correspond to the results from the reported multivariable analysis?

**Table 1 jpm-12-01682-t001:** Characteristics of included studies.

		Patients	Predictors			Outcomes		Analysis	
Authors (Year)	Country	Data Source	Training Data Set for Development (Training Set)	Test Data Set for Validation (Test Set)	Predictors/Variables	Level	Outcomes	Algorithms	Performance Metrics
Development and Validation studies									
Krois et al. (2019) [17]	Germany	Two cohorts of periodontal patients in two universities (Kiel & Greifswald) in Germany, 627 patients, 11,651 teeth	From data source, six specific cohorts were used for training in “Hold-out validation”.	From data source, six specific cohorts were assessed for validation in “Hold-out validation”.	4 patient-level outcomes, 6 tooth-level	tooth	tooth loss during SPT	RFO, XGB, DTC, logR	AUC, sensitivity, specificity, the no-information rate
Cui et al. (2021) [18]	China	Cohorts of prosthodontic patients in Chinese University (Peking), 3559 patients, 26,005 teeth	From data source, randomly selected from data source (18182 teeth) in “Hold-out validation”.	From data source, randomly selected from data source (7823 teeth) in “Hold-out validation”.	34 oral outcomes	tooth	tooth extraction/retention	DTC, AdaBoost, GBDT, LightGBM, XGB	AUC, sensitivity, specificity, accuracy, precision, F1
Cooray et al. (2021) [19]	Japan	Japanese community cohort, 19,407 patients aged 65 and older	From data source,10-fold cross validation was used for model development.	From data source, 10-fold cross validation was used for model validation.	14 oral and socioeconomic variables	patients	Tooth loss, Tooth loss number category	RFO, XGB	AUC, accuracy, precision, F1
Elani et al. (2021) [20]	USA	National Health and Nutrition Examination Survey (NHANES) from 2011 to 2014	NHANES 2011 to 2012 (n = 5,864)	NHANES 2013 to 2014 (n = 6,113)	(1) 28 items; socioeconomic characteristics, routine dental care, and chronic medical conditions, (2) the number of decayed teeth, periodontal disease, age, gender, race.	patients	edentulism, having fewer than 21 teeth, missing any tooth	logR, RFO, LightGBM, XGB, artificial neural networks.	AUC, accuracy, sensitivity, specificity, F1, positive predictive value, negative predictive value, the harmonic mean for sensitivity and specificity for each predictive model.
Development studies									
Lee et al. (2022) [21]	USA	Electric data at Harvard Medical School pf 94 patients with 2539 teeth	All of the data source	NA	17 parameters including medical and dental conditions	tooth	tooth prognosis ranking 1 to 5 decided by 16 dentists (ModelA), and 13 prosthodontists (ModelB)	XGB, RFO, DTC	accuracy

RFO: random forest; XGB: extreme gradient boosting; DTC: decision tree classification; logR: logistic recession; LightGBM: light gradient boosting machine; AdaBOOST: adaptive boosting; AUC: area under the curve; F1: F1 score.

**Table 2 jpm-12-01682-t002:** Main results and conclusions of included studies.

		Results			Conclusion
Authors (Year)	Country	AUC	Accuracesy	Summary	
Development and Validation studies					
Krois et al. (2019) [17]	Germany	In Scenario1, RFO:0.84, XGB:0.84, DTC:0.76, logR:0.8	In Scenario1, RFO:0.92, XGB:0.91, DTC:0.91, logR:0.92	More complex models (RFO, XGB) had no consistent advantages over simpler ones (logR, DTC).	None of the developed models would be useful in a clinical setting, despite high accuracy. During modeling, rigorous development and external validation should be applied and reported accordingly.
Cui et al. (2021) [18]	China	In triple classification, DTC:0.931, AdaBoost:0.924, GBDT:0.966, LightGBM:0.975, XGBoost:0.969	In triple classification, DTC:0.915, AdaBoost:0.895, GBDT:0.916, LightGBM:0.921, XGBoost:0.924	The XGBoost outperformed the other 4 algorithms.	A clinical decision supportmodel for tooth extraction therapy achieved high performance in terms of decision-making derived from electronic dental records.
Cooray et al. (2021) [19]	Japan	In random oversampling analysis (with/without tooth loss = 1), RFO:0.827, XGB:0.905	In random oversampling analysis (with/without tooth loss = 1), RFO:0.827, XGB:0.906	XGB outperformed RF model, and predicted the tooth loss with a satisfactory level of accuracy.	In addition to oral health related and demographic factors, socioeconomic factors are important in predicting tooth loss.
Elani et al. (2021) [20]	USA	For edentulism; logR:0.865, RFO:0.885, LightGBM:0.884, XGB:0.887, artificial neural networks:0.877. For having fewer than 21 teeth, logR:0.872, RFO:0.876, LightGBM:0.877, XGB:0.883, artificial neural networks:0.881. For missing any teeth, logR:0.819, RFO,:0.827, LightGBM:0.819, XGB:0.832, artificial neural networks:0.831.	For edentulism; logR:0.837, RFO:0.843, LightGBM:0.827, XGB:0.838, artificial neural networks:0.822. For having fewer than 21 teeth, logR:0.819, RFO:0.817, LightGBM:0.825, XGB:0.815, artificial neural networks:0.826. For missing any teeth, logR:0.769, RFO:0.770, LightGBM:0.739, XGB:0.740, artificial neural networks:0.772.	XGB had the highest performance in predicting all outcomes.	Our findings support the application of machine-learning algorithms to predict tooth loss using socioeconomic and medical health characteristics.
Development studies					
Lee et al. (2022) [21]	USA	NA	For Model-A, XGB:0.689, RFO:0.8312, DTC:0.8413. For Model-B, XGB:0.6687, RFO:0.7421, DTC:0.7523.	DTC had the best accuracy among the three methods. Model-A indicated a higher accuracy than Model-B for al models.	AI-based machine-learning algorithm will be a helpful tool to determine tooth prognosis in consideration of the treatment plan.

RFO: random forest; XGB: extreme gradient boosting; DTC: decision tree classification; logR: logistic recession; LightGBM: light gradient boosting machine; AdaBOOST: adaptive boosting; AUC: area under the curve.

## Data Availability

Not applicable.

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
