# Peer review of "Machine Learning in Predicting Tooth Loss: A Systematic Review and Risk of Bias Assessment"

_jpm, 2022, doi:10.3390/jpm12101682_

Round 1

Reviewer 1 Report

This is a systematic review aimed to evaluate the risk of bias in prognostic prediction models of tooth loss that use machine learning. The reviewer found this manuscript informative, however, there are few concerns on this manuscript.

1. Tooth loss as the outcome measure

The authors cited prognosis systems from Hirschfeld and Wasserman and McGuire which heavily emphasized on periodontal involvement. However, within the selected 5 literatures, the cause of tooth loss as well as its definition were not explained and generalized the results from selected literatures may have great bias for the results of the systematic review.

For example, if there is any of such “natural tooth loss” vs extracted due to decay/periapical pathology vs periodontally compromised? Does the definition of tooth loss include the extractions performed with the reason other than hopeless prognosis (such as prosthetic reason etc)?

2. Inclusion of one development study

Including Lee 2022 study may add more bias on the results.

3. Limitation of the current manuscript

Please discuss the limitation of the current literature in detail.

Author Response

Dear reviewer,

Thank you for your fruitful comments. We revised the manuscript following your comments.

#1. Tooth loss as the outcome measure.

Agreed.

Including tooth loss outcome with any reason must be one of the limitation  of the present review. I added the description as the limitation. (line 35-355)

2. Inclusion of one development study

Agreed.

Although  ROB (risk of bias) of Lee 2022  is judged as critically low, this study was included due to the fulfillment of inclusion criteria. The description was added in the discussion (line 332-337).

3.Limitation of the current manuscript

Thank you for your instruction.

I added comments on limitations in discussion (line 351-358).

I appreciate all of your help.

Sincerely,

Akira Hasuike.

Reviewer 2 Report

Dear authors,

this is a good article about a new topic regarding the era of digital dentistry. Machine learning algorithms may aid the dentist in the processes of diagnosis and prognosis especially in this prevalent scope.
The article is well-written, materials and methods are well explained and statistics is appropriate.
However, some corrections are needed in order to improve the quality of your article. Please, see below the recommended corrections. 

- In the forehead of the manuscript you can change the written "Type of paper (Review)" in "Review" only

- Definition of AUC in the abstract must be declared in order to clearify its significance at the beginning of the article

- In the introduction section, phrases from line 85 to 91 have to be excluded and inserted in the other subsequent sections of the manuscript

- in the discussion, a section of limits of this study is necessary

- I would emphasise the modifications and improvements needed to augment the reliability of studies taking into account machine-learning algorithms to predict tooth loss 

Author Response

Dear reviewer,

Thank you for your fruitful comments. We revised the manuscript following your comments.

#1. change the forehead from "Type of paper (Review)"  to "Review" 

Thank you for your instruction. I changed.

#2. Definition of AUC in the abstract 

Thank you for your instruction. 

I added the description of AUC in both the abstract(line25-28) and main text(line173-178).

3. In the introduction section, phrases from line 85 to 91 have to be excluded and inserted in the other sections.

Thank you for your instruction. 

I deleted these lines.

4. in the discussion, a section of limits of this study is necessary

Thank you for your instruction.

I added comments on limitations in discussion (line 351-358).

5. emphasise the modifications and improvements needs

Thank you for your instruction.

I added comments on modifications and improvements in discussion(line291-293) and conclusion (line 369-370).